

# Deep gradient reinforcement learning for music improvisation in cloud computing framework

Fadwa Alrowais[1], Munya A. Arasi[2], Saud S. Alotaibi[3], Mohammed Alonazi[4], Radwa Marzouk[5] and Ahmed S. Salama[6]

[1] Department of Computer Sciences, College of Computer and Information Sciences, Princess Nourah bint Abdulrahman University, Riyadh, Saudi Arabia
[2] Department of Computer Science, Applied College, King Khalid University, RijalAlmaa, Saudi Arabia
[3] Department of Computer Science and Artificial Intelligence, College of Computing, Umm Al-Qura University, Mecca, Saudi Arabia
[4] Department of Information Systems, College of Computer Engineering and Sciences, Prince Sattam bin Abdulaziz University, Al-Kharj, Saudi Arabia
[5] Department of Information Systems, College of Computer and Information Sciences, Princess Nourah bint Abdulrahman University, Riyadh, Saudi Arabia
[6] Department of Electrical Engineering, Faculty of Engineering & Technology, Future University in Egypt, New Cairo, New Cairo, Egypt

Corresponding author
Radwa Marzouk,
rmmarzouk@pnu.edu.sa

## ABSTRACT

Artificial intelligence (AI) in music improvisation offers promising new avenues for developing human creativity. The difficulty of writing dynamic, flexible musical compositions in real time is discussed in this article. We explore using reinforcement learning (RL) techniques to create more interactive and responsive music creation systems. Here, the musical structures train an RL agent to navigate the complex space of musical possibilities to provide improvisations. The melodic framework in the input musical data is initially identified using bi-directional gated recurrent units. The lyrical concepts such as notes, chords, and rhythms from the recognised framework are transformed into a format suitable for RL input. The deep gradient-based reinforcement learning technique used in this research formulates a reward system that directs the agent to compose aesthetically intriguing and harmonically cohesive musical improvisations. The improvised music is further rendered in the MIDI format. The Bach Chorales dataset with six different attributes relevant to musical compositions is employed in implementing the present research. The model was set up in a containerised cloud environment and controlled for smooth load distribution. Five different parameters, such as pitch frequency (PF), standard pitch delay (SPD), average distance between peaks (ADP), note duration gradient (NDG) and pitch class gradient (PCG), are leveraged to assess the quality of the improvised music. The proposed model obtains $+0.15$ of PF, $-0.43$ of SPD, $-0.07$ of ADP and $0.0041$ NDG, which is a better value than other improvisation methods.

# INTRODUCTION

Music is claimed to be the universal language of people worldwide. The beneficial effects of music on mental health have been confirmed by numerous research, including more recent studies such as *Ghatas, Fayek & Hadhoud (2022)*, and *Dua et al. (2020)*. Additionally, there is evidence in the literature that machine learning algorithms have used this phenomenon (*Modrzejewski, Dorobek & Rokita, 2019*). Indian classical music, which has been around for thousands of years since the Vedic era, is especially claimed to be beneficial for several medical ailments (*Hewahi, AlSaigal & AlJanahi, 2019*). But strong music comes from years of intense practice under the guidance of a seasoned professional artist.

Machine learning-based audio analysis has reached areas including detecting speech and auditory identification of scenes (*Carnovalini & Rodà, 2019*). Machine learning will eventually replace human listeners for various functions, including music instruction. Recently, *Dong et al. (2018)* published their work on changing the melodic framework of Indian classical music. Hindustani classical music is a specific genre of music with antecedents in Indian origin. It is a diversified and traditional type of musical art. The fundamental component of Hindustani classical music is known as raga.

A raga is a subject in music or a melodic framework made up of a particular group of notes on a scale, which are then arranged in tunes with harmonic patterns. Its distinctive rhythms, adornments, and frequencies are a foundation for improvisation. Technically speaking, experts define range as a set of rhythmic elements and a method for developing them (*Dong & Yang, 2018*). Hindustani classical music emphasises the intervals between notes as much as the real sounds. While playing a melodic framework, a musician may stick to the same notes or add more emphasis to specific scale degrees to create a mood exclusive to the melody.

Every melodic framework may be written on a scale and comprises five or more notes. Numerous musical frameworks have a standard scale and incorporate various improvisations, including the time to place and intervals between notes. These improvisational patterns are hard to pinpoint, poorly documented in any format, and only a select group of musicians who have mastered them know them. Because of the irregular rhythmic separation and harmonic diversity, identifying the particular framework (raga) is a tough task (*Wu & Yang, 2020*).

Not many artists have mastered it, yet various musical themes or ragas can be built around the same scale with multiple hard-to-remember improvisations. There isn't even a structure that clearly defines these improvised rhythms. This work suggests a method for determining which melodic frameworks a piece of music abides by using deep learning techniques. Deep learning is utilised to discover melodic frameworks based on audio parameters, including pitch, tone, and scale, and to incorporate improvisations.

Music improvisation can be produced by hand or, using various methods, by a computer. Music can be improvised *via* machine learning methods (*Frieler & Zaddach, 2022*). There's a new phrase for the type of music that describes itself as continually evolving and changing due to systemic influences. Many fields are already using machine learning to better their practices. It has always been aimed to digitalise tasks and make things easier as technology

advances. Even though machine learning has been successfully applied in every sector, it can be used greatly in most artistic disciplines, including music. To retain variety in the music, the rhythmic notes in the musical composition need improvisation (*Korkmaz, Boyacı& Tuncer, 2019*). It might be advantageous to use machine learning to resolve this issue.

New musical compositions are always in demand for various applications, such as lift soundtracks, video game soundtracks, TV show commercials, and creative expression. As a result, to train a model, it must be given a portion of a music file as input and be able to generate the remainder of the file (*Dighe et al., 2013*). The output that is produced ought to resemble the dataset that was provided as input. This compact technique that has been presented uses machine learning to deliver original music content in addition to genre-based music generation. Determining some of the measures allows one to gauge the quality of the generated music. The music genre in this work is classical music, and the songs are chosen as compositions by different musicians. After being gathered from open-source datasets, they should be transformed into the necessary MIDI format. It is prudent to ensure that the datasets selected are similar to the desired final output and do not contain these discrepancies. This suggested flexible model combines genre-specific improvisation with reinforcement learning to produce novel music material.

The main contributions of this work are,

1. To provide a deep learning-based technique for detecting the melodic framework using bi-directional gated recurrent units architecture.
2. To demonstrate the improvisation of the identified melodic framework using a deep gradient-based reinforcement learning technique.
3. To render the improvised music in MIDI format and analyse the quality of the outcome using various performance measures.

The remainder of the paper is organized as follows: 'Related Works' discusses recent works related to the application of machine learning or deep learning techniques for music generation or improvisation. 'Proposed Methodology' elaborates on the methods employed in the proposed research for identifying melodic frameworks, improvising musical notes, and rendering them in MIDI format. 'Results and Discussion' presents the results obtained by applying the deep learning techniques to the Bach Chorale dataset by implementing it in the cloud environment. 'Conclusion' concludes the present research.

## RELATED WORKS

In this section, the state-of-the-art works utilised for music improvisation, as well as generation in the literature, are discussed. The process of automatically creating music on computers by leveraging deep learning network topologies is deep music production learning architectures substantially outperformed manual feature extraction techniques in ImageNet workloads. Subsequently, deep learning has gained popularity and steadily expanded into an area that has developed quickly. Naturally, as a subject of study that has been active for decades, music production and improvisation have drawn the interest of several researchers. Deep learning algorithms are currently the most widely used technique in the music production industry.

The authors discuss a mixed deep learning strategy in *Joseph & Vinod (2017)* for estimating the musical difficulty of symbolic piano music. This study uses several levels and divides the piano roll to build a sophisticated neural network model. When creating the models linked to artistic creations, several features are considered. This work uses deep neural network architectures (*Samsekai Manjabhat et al., 2017*) to execute a computer-generated version of traditional music based on musical compositions. To produce music, this study employed two different neural network architectures.

The models' precision and error are used to gauge their performance. Additionally, the input fed during the training phase is optimised by this procedure. To improve the accuracy of the music production, the researchers discussed a method for sheet music generation based on autoencoders. This study changes the harmonic prediction and source segmentation modules to improve the models. This work employs a variety of deep learning models, including multilayer perceptrons and radial basis function networks. In this case, the total quantity of resources has been divided to improve accuracy.

A unique recurrent neural network (RNN) based model was created by *Kritsis et al. (2021)* to help the network remember and fetch data in the series. This model was employed in music production in *Madhok, Goel & Garg (2018)* for the first time, which used a brief recording to improvise traditional blues tunes with a decent tempo and coherent framework. The Restricted Boltzmann machine model was proposed in *Hsu & Chang (2021)* for music improvisation, and although it outperforms the conventional harmonic music composition model in several datasets, capturing the music structure with prospective reliance remained difficult. The researchers suggested the Harmony RNN model in *Degaonkar & Kulkarni (2018)*, which enhanced RNN's capacity to learn persistent patterns. Eventually, a unique RNN model enabling user-defined temporal restrictions was introduced in *Jamshidi, Marghitu & Chapman (2021)*.

Strong computational models like Transformers, generalised neural networks and variational autoencoders have steadily surfaced as deep learning technologies advance. *Chen, Xiao & Yin (2019)* and *Moysis et al. (2023)* presented a deep convolutional GAN (DCGAN) for music generation. It is capable of performing music synthesis from audio features. Jin et al. (2020) considered various deep learning approaches to music generation, focusing on the application of deep generative adversarial networks (GANs) and convolutional networks (CNNs) to music creation.

The model presented in *Pendyala (2020)* is thought to be the first to produce multi-track harmonic music. For the first time, *Rahman et al. (2021)* successfully used RNN-based GAN to generate music by *Moreira, Furtado & Nogueira (2020)* combining reinforcement learning technologies. Transformer models have recently demonstrated their enormous musical potential. Transformer was used to successfully create music with a long-term structure for the first time. *Surana, Varshney & Pendyala (2022)* presented a pre-training technology based on transfer learning and suggested using a Transformer to create multi-instrument music. *Liu & Tu (2020)* using the machine language model (*Mao, Shin & Cottrell, 2018*) as the progression model and an innovative music representation technique.

Studies on deep learning-based performance generation are scarce in the music industry (*Bell, 2011*). Most performance-generating models are straightforward temporal-based models in contrast to different intricate performance improvisation approaches. Other studies improvise melodic polyphony with emotive tempo and rhythm using DNN models; these models resemble structural models more than reactive performance simulations that take score input. A recent work by *Hassani & Wuryandari (2016)* proposed a DNN-based emotive drumming modelling system, apart from piano playing. The scarcity of datasets is a barrier to deep learning-based performance improvisation. The datasets should include human musicians' scores and corresponding performances to improvise creative performances from music scores. Additionally, for model training to be effective, scores and performance pairings must be harmonised at the chord level.

*Yin et al. (2023)* assesses many deep-learning algorithms for music production, emphasising their drawbacks and possible enhancements. The authors present a comprehensive comparative analysis that highlights the surface-level developments in automatic music generation and suggests that a more thorough integration of creative processes and musical theory could improve the efficiency of these algorithms.

These are facilities that provide shared computing resources over the Internet. Cloud data centres host and manage VMs, offering scalable resources to meet the varying demands of multiple tenants (*Sun et al., 2015*; *Sun et al., 2018c*). Cloud computing involves using a network of remote servers hosted on the Internet to store, manage, and process data rather than a local server or a personal computer (*Shang & Luo, 2021*; *Jiang et al., 2021*). VR platforms can enhance the quality of education by providing students with interactive and engaging learning experiences (*Liu et al., 2024*). Adapting to changing user patterns and mobility in real-time can be challenging, requiring advanced predictive algorithms and responsive network systems (*Li, 2024*; *Ban et al., 2023*). Applying advanced training techniques, such as dropout for regularisation (to prevent overfitting) or techniques like gradient clipping to handle exploding gradients, can also improve the performance of LSTMs in text filtering tasks (*Dang et al., 2023*; *Li et al., 2020*).

The study presents "LyricJam Sonic", a generative system (*Johnson, Rodríguez-Fernández & Rebelo, 2023*) for improvisation and real-time music composition. This work is noteworthy because it examines the relationship between artificial intelligence (AI) and live performance, highlighting how AI may foster more musical creativity in these contexts. The system is a noteworthy addition to the field of artificial intelligence in music because of its capacity to dynamically interact with musical inputs. *Hu (2023)* investigates how to use reinforcement learning algorithms for Internet of Things (IoT)-specific music improvisation and arranging in sensor networks. Hu's work is groundbreaking because it explores the use of AI in music to the Internet of Things (IoT) and provides insights into how these technologies might work together to produce new musical experiences and capabilities.

*Moysis et al. (2023)* thoroughly analyses deep learning techniques used in music signal processing. It provides a thorough summary of the state-of-the-art at the moment by covering a variety of methodologies and their applications. For academics interested in

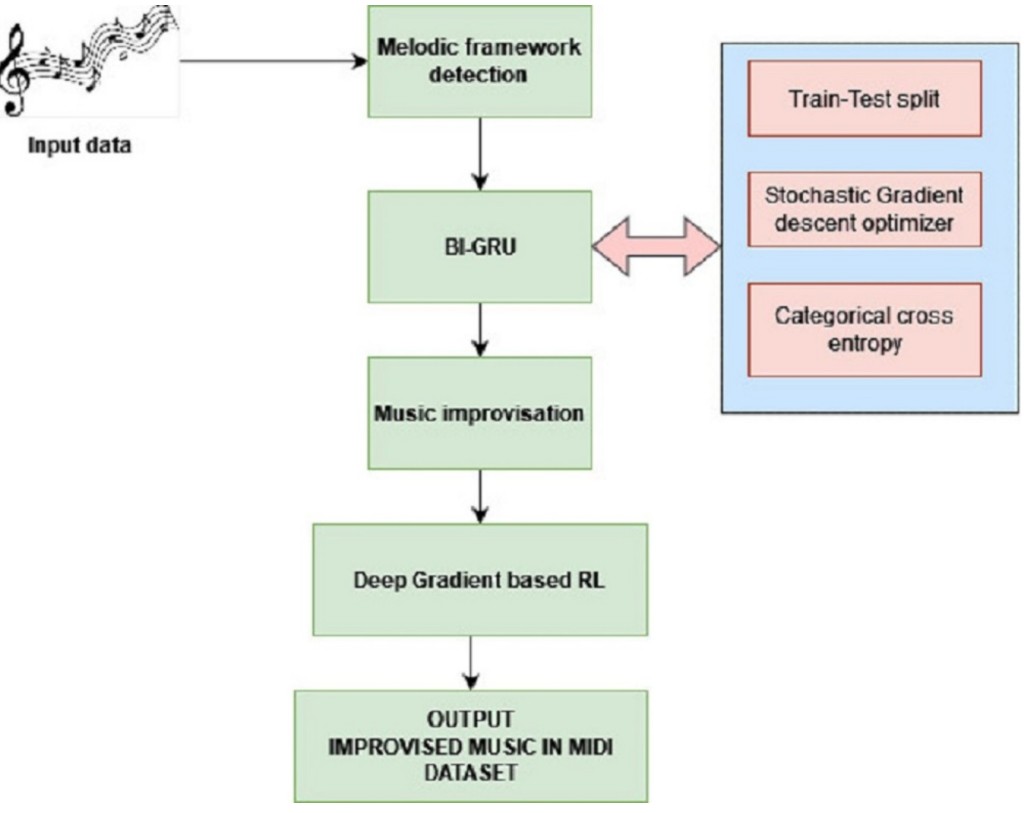

**Figure 1** Proposed architecture.

the nexus between artificial intelligence and music technology (*Dai, 2023*), the writers' discussion of difficulties and potential solutions makes it a helpful tool.

The recommendations are presented to the user through the smartphone interface. The system continuously learns and adapts to the user's responses, refining its recommendations over time (*Shen et al., 2022*; *Xiao et al., 2021*). Messages are forwarded from vehicle to vehicle, prioritising vehicles expected to have the best chance of successfully delivering the message (*Sun et al., 2018b*; *Sun et al., 2018a*). Identifying and summarising the main points of large documents by understanding the semantic relatedness between sentences (*Gu et al., 2024*; *Ding et al., 2023*). The system uses a neural network designed to handle and integrate these different types of data (*Pan et al., 2023*; *Pan et al., 2024*). A combination of deep learning and reinforcement learning. Deep learning involves neural networks with many layers (deep networks) that can learn from large amounts of data (*Zhu, 2023*; *Qin et al., 2024*).

# PROPOSED METHODOLOGY

The research in this work proposes improvising the music compositions by identifying the melodic frameworks and rendering them in MIDI format after improvisation. The architecture of the proposed system is presented in Fig. 1.

Scalability, both vertical and horizontal, is fundamental to cloud platforms. Horizontal scalability entails adding new instances or servers (scale-out), whereas vertical scalability permits expanding the capacity of already existing servers (scale-up). Numerous cloud services have auto-scaling capabilities, automatically modifying resources in response to demand. This is crucial for systems with unpredictable usage patterns to ensure peak performance and off-peak cost reduction. Cloud providers efficiently manage resources by utilising modern optimisation techniques. Because they can isolate dependencies and are lightweight, containers are perfect for microservices design. This makes it easy to scale out portions of an application that need more resources without expanding the complete program since it enables the independent scaling of its component pieces.

## Dataset collection

An essential part of the learning process is the data that will be entered into the model. When these data are improperly selected, a model may yield unfavourable outcomes. Even the choosing process needs to be done with caution. The chosen data should be pertinent to the scenario and problem description. The first stage is choosing songs that are comparable to each other, which means they employ the same instruments and are in the same genre. The outcome of selecting tracks that are not related is unappealing music. Occasionally, the datasets that are gathered may come from different sources. It is important to carefully alter them into a comparable format with the same quantity, kind, values, and names of characteristics. This proposed approach focuses on the classical music genre, which is composed by a variety of performers. These are gathered from several public datasets and should be transformed into the necessary MIDI format.

The harmonised chorales of J.S. Bach that make up the Bach Chorales collection are renowned for their melodic coherence and structural consistency. This homogeneity is essential because it offers a reliable foundation for training machine learning models. Bach is well known for his compositions' complex harmonic progressions and melodic lines. This intricacy provides an extensive dataset full of examples of complicated ideas in advanced music theory. Bach Chorales are typically offered in MIDI format, which is perfect for the needs of this project. MIDI files include comprehensive timing, duration, and pitch information for training music-generating models. This study intends to improve melodic framework identification and improvisation to improve music creation. Bach Chorales provide a great opportunity to investigate and play around with these frameworks because of their distinct and unambiguous melodic structures. This research uses a wealth of classical music compositions consistent in style and complexity and appropriate for deep learning applications in the music industry by choosing the Bach Chorales dataset.

Dataset 2: This analysis section uses the MAESTRO dataset (*Hawthorne et al., 2018*). It is a collection of 1,276 MIDI data recordings of virtuoso piano performances, totalling 7.04

million notes, made with Yamaha Disklaviers. In contrast to the MIDI data gathered for KernScores, this data features dynamic and expressive timing.

### Distinctive music notes selection

Because the dataset is made up of different songs, there are more distinctive note combinations that must be predicted while creating music. Every note that needs to be anticipated is comparable to a class in a machine learning classification issue. To classify the inputs in a categorization problem, they are taken and modified through several hidden layers. Before training the model, the number of concealed layers, connections in every layer, and the layer resulting from training must all be determined. Because of this, it is crucial to understand how many nodes—equal to the number of classes—are present in the layer that produces the output.

### Music sequence segmentation

While the songs in the dataset can differ in length, the model only accepts a limited number of inputs determined by its architecture. This could make using them as inputs problematic. Poor outcomes will arise from improper feeding of the datasets. To address this issue, the dataset is split into same-sized sequences, which are then used to train the model. Each of these sequences is fed into the model for a single instance. Subsequently, the model teaches itself to anticipate which note will come next in the input sequence. The segment size is a parameter that needs to be carefully set because it will determine the number of inputs in the model design.

### Bi-directional gated recurrent units

Bi-directional gated recurrent unit (Bi-GRU) is implemented to detect the melodic frameworks in the proposed approach. This deep learning-based model is developed on the audio attributes to determine the chord progression of a submitted music segment. The bidirectional GRU model is trained using the characteristics extracted from the music files. GRU is a prevalent option for problems involving periodic and series-to-series categorisation. This unique type of RNN can take feature dependencies into account over an extended training period. This model includes an attention state that aids in retaining or losing details, such as one's capacity to comprehend music.

The attention preserves significant aspects of the music that are necessary for later consideration. Because of this feature, Bi-GRU performs well when dealing with serialised data, such as audio signals. Dividing the concealed state into two halves is an essential part of Bi-GRU. The first portion is stored for a long time whereas the second component is reserved for only a brief period. Similar to how the human mind processes music, some information about the audio features is preferentially devoured; others are retained, while others are disregarded or lost.

One component of the model architecture is a progressive network. The input layer is connected to the Bi-GRU layer to lessen bias, which is subsequently attached to the dropout layer. It is possible to add more layers to improve the functionality and feature extraction of the Bi-GRU model. Including a few dropout layers in between is crucial to ensure the model can learn correctly. To gather all the data from all the nodes into a single layer that

can eventually be connected to the output layer, the model's last layer is compressed into one and linked to an intense layer. The output layer for the last calculations requires an activation function.

Activation functions are used at this process stage to determine a node's output in neural networks given a certain set of inputs. The activation function of digital networks is achieved by using a chip circuit dependent on the binary input and using the neural networks of the linear perceptron. Predicting which note will occur next is likewise a multiclass classification because the number of notes in the songs is always high and the number of categories is also considered. Thus, in this specific model, the activation function is softmax. Another excellent option for the problem's activation function is the tanh function.

The equal-length sequences obtained during the data preprocessing stage train the developed model. To anticipate the following note to be played, the entire dataset, separated into many sequences, is run through these sequences. The difference between the expected and real notes is computed to determine the loss. The various weights and biases in the model are now adjusted using the computed loss to make the subsequent prediction more accurate the next time. By doing this, the loss is reduced for the future.

## Deep gradient-based reinforcement learning

This work uses deep gradient-based reinforcement learning for real-time music improvisation by optimising the rewards. A policy that improves the anticipated prospective total reward is learned through reinforcement learning. The prospective reward can be $R_\theta S$. The adjusted discount at any instant of time k is represented in Eq. (1),

$$R_k = \sum_{a=0}^{K} \mu^a r_{a+k}. \tag{1}$$

In the above equation, $r_k$ denotes the standards of the generated next note. $\mu$ represents the rate of discount that is applied. The model continually learns the actions that must be performed and its corresponding value function. The action to be taken by the agent is represented as $\theta_{\alpha_x}(x_k|h_k)$ and the function is denoted as $F_{\alpha_x}(h_k)$. Accordingly, the agent is made to proceed with the actions $x_k$ for the different states $h_k$. The value of the discounts generated is computed using the Eq. (2):

$$d^\delta = R\left[\sum^{K_{k=0}} \nabla_{\alpha_x}\log\theta_\alpha(x_k|h_k)\sum^{K_{n=0}}(\delta\beta)^f \gamma^F_{k+f}\right]. \tag{2}$$

In Eq. (2), the value of $\gamma^F_k$ can be determined by using the representation in Eq. (3) which specifies the resultant of the time variation in the value function,

$$\gamma^F_k = x_k + \delta F(h_{k+1}) - F(h_k). \tag{3}$$

The slope value of the reward function is computed using the formulation in Eq. (4),

$$R_s = R\left[\sum_{k=0}^{K} \nabla_{\alpha_x}(H(||F_{\alpha_x}(h_k) - R_k||^2))\right]. \tag{4}$$

The training procedure of the DGBRL method alternates between the creation and reward stages. During the creation phase, manual and machine-generated equivalent tokens are successively produced by the creation agent in a given state, $h_k$. Following the musical improvisation, the reward agent calculates the reward for every move at every step. Subsequently, these rewards are utilised to compute $R_k$ and are employed to update the reward function $F_{\alpha_x}(h_k)$ and the creation agent $\theta_{\alpha_x}(x_k|h_k)$.

Further, the model adapts as it processes and learns from a new token, either by absorbing the new data or creating a new category. Absolute mobility is the ability of the model to measure the precise change in the upcoming tokens. The DGBRL's inherent reward R is calculated as the amount of the shift observed between the state before the new input data ($h_{k-1}$) and the modified state ($h_k$) using the representation in Eq. (5):

$$R = \sqrt{(h_k^x - h_{k-1}^x)^2}. \tag{5}$$

To avoid the input values that contribute minimally towards the improvisation of the musical notes, a minimum limit is being set on the reward value based on the endurance e as shown in Eq. (6):

$$R_l = \frac{e}{|R - e| + e}. \tag{6}$$

Consequently, input tokens that yield negligible changes are considered uninteresting and unrewarding. The more unpredictable and less comparable an input is to previously identified patterns, the less rewarding the difference is from it. Using the above equation, the DGBRL's music improvisation phase can guarantee that the most satisfying decision is always taken by selecting the most rewarding following stimuli.

The problem of music improvisation can be defined in a finite amount of time, for a certain number of steps, evaluating all potential future inputs, provided that pitch region and rhythm are periodic. Music improvisation at every phase entails figuring out what would happen if you observed each option. When the model runs consistently, it generates the option that will yield the greatest reward, plays the corresponding pitch, and records the outcome. Theoretically, this leads to investigating all local maxima; however, as the points become less significant with every presentation, the agent is never trapped.

## Generation of MIDI format

The music file's output is extracted and must be converted to MIDI format. Since the production of music generation is not a value that can be compared and verified for correctness, it cannot be directly evaluated. It is necessary to hear the music to determine whether or not it is pleasing to the ear and whether it generates the required music in the appropriate genre with the proper instruments. This is accomplished by converting the output file to MIDI format using online converters or open-source programs. This output can now be transferred and listened to like any other music file and can be played with a music player.

# RESULTS AND DISCUSSION

## Experimental setup

Owing to its ease of use, uniformity, device autonomy, and availability of excellent tools and software for deep learning (DL) and machine learning (ML), Python was chosen as the programming language for this research setup. Python 3.11.5 is the version employed. The backend framework used in this proposed system was TensorFlow 2.10.0. Pandas, sci-kit-learn, and Keras are the three main machine-learning libraries adopted.

The dataset is split into train, test, and validation sets using the standard train_test_split technique in the Scikit-learn module. Twenty percent of the train data went to the validation split, and twenty percent went to the test split, as is customary. After that, the BiGRU model was trained for 1,000 epochs, leveraging 75% dataset. The model was prepared using the Stochastic gradient descent optimiser with a learning rate of 0.001 and a loss function of categorical cross-entropy.

The scope of the chosen problem and model efficiency determines which loss function and optimiser to use in the model implementation. The problem in the current research is a multi-class classification problem, which is ideally suited for the use of categorical cross-entropy. It should be mentioned that the goals are also expressed as integers. Since the encoded data takes up significantly less space than one-hot encoding, the target is minimal. The stochastic gradient descent technique is more suitable based on the dynamic approximation of the initial and higher-order instances. It provides an optimisation approach that can handle scant slopes on chaotic issues by combining the best features of the RMSProp along with Adam algorithms.

## Model implementation

Model deployment is one of the most important phases of a machine-learning application. The model must be highly efficient and extensible to support thousands of users. As a result, the cloud, along with containerization architecture, was used to deploy the proposed system. In any case, a significant portion of computing in recent years has occurred in the cloud. The deployment expands flexibly and enables the system to be used by a small number of or millions of users. In this research, this system only recognises the uploaded music's melodic structure but improvises the raga's vicinity and converts it to MIDI format.

The use of the cloud and containerisation services has significantly advanced over time into sophisticated DevOps technology, which handles the distribution of load to satisfy unpredictable demands from global clients. Thus, DevOps in a solid cloud infrastructure is employed to simultaneously cope with even millions of requests. Docker employed technology to deliver container software by utilising virtualisation at the Operating System level to achieve containerisation. With Docker, developers can quickly pack, deploy, and operate any programme as an independent, compact container that can function almost anytime. Deploying Docker containers in a cloud environment is also simple.

The orchestration of containers was done using Kubernetes. Kubernetes is a freely available container management technology that allows an extensible web hosting framework for cloud apps to run. It provides mobility and quicker, more manageable deployment timeframes. The prediction model's containers were implemented on the

**Table 1  Accuracy analysis of traditional models with two datasets.**

| Deep learning technique | Accuracy (%) (dataset 1) | Accuracy (%) (dataset 2) |
|---|---|---|
| RNN | 79.6 | 79.02 |
| LSTM | 82.3 | 81.21 |
| Bi-LSTM | 87.5 | 88.15 |
| ResNet | 84.7 | 84.56 |
| DenseNet | 91.4 | 90.87 |
| Bi-GRU | 95.6 | 95.41 |

Google Cloud platform's Kubernetes group. Every node comprises several hosts, each of which may hold one or more containers.

## Datasets

The dataset used in the implementation is the Bach Chorale dataset, a univariate sequential time series dataset with six features. It primarily comprises Single-line melodies of 100 Bach chorales. The six different features of the dataset are start-time, pitch, duration, key signature, time signature and fermata. Each chorale consists of four sections that are monolingual and univocal. The dataset utilised in the implementation can be accessed using the following link: https://archive.ics.uci.edu/dataset/25/bach+chorales.

## Experimental results

Initially, experiments were conducted to detect the melodic framework in the input audio songs, which were further used for music improvisation using deep gradient-based reinforcement learning. Bi-GRU was employed to identify the raga in the songs. The audio features were extracted, and the number of raga classes in specific notes was also analysed. To conclude the performance supremacy of Bi-GRU, a few existing methods were also considered for performance comparison. The conventional deep learning methods such as RNN, long short-term memory networks (LSTM), bi-directional long short term memory networks (Bi-LSTM) and transfer learning techniques such as ResNet and DenseNet models were utilised for analysing the efficiency of Bi-GRU in detecting the melodic frameworks. Table 1 shows the accuracy exhibited by each model in detecting the raga in the audio input.

It takes significant computational power to train and implement a complex music creation model in a cloud environment using Bi-GRUs and deep reinforcement learning. Because of the complexity and scale of the neural network designs involved, such as transformers or hybrid models that combine CNNs and RNNs, the model initially requires high-performance GPUs for effective training. These GPUs speed up matrix calculations, which are essential to deep learning. Moreover, big datasets and the maintenance of many model instances during training stages require sufficient RAM, especially for batch processing and data augmentation tasks.

It can be observed from Table 1 that the RNN technique produced an accuracy of 79.6% in particularly detecting the ragas in the music. The LSTM method was 82.3% accurate in detecting the melodic ragas. Bi-LSTM exhibited higher accuracy than LSTM

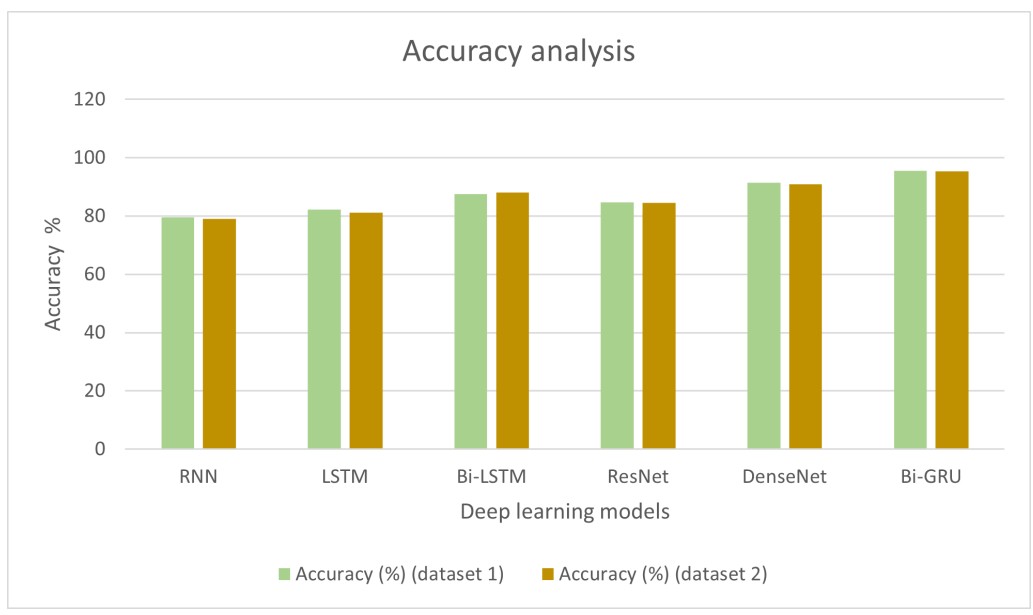

**Figure 2** **Accuracy analysis of different deep learning techniques.**

with 87.5%, which was also higher than the efficiency of the ResNet model with 84.7%. DenseNet showed higher efficiency than the other models considered for comparison, which is 91.4%. However, it was conclusive from the accuracy assessments that the Bi-GRU technique employed in the present research can detect the ragas with 95.6% accuracy, which is the highest among the other deep learning techniques compared. The results of the analysis are presented graphically in Fig. 2.

Once the melodic frameworks are identified, the next step is to improvise the music using the reward model of the DGBRL method. The agent involved in the implementation employs five different reward models trained previously for 100 epochs at differing learning rates of 0.01, 0.001, 0.05, 0.005 and 0.02. In the DGBRL technique, there are two learning phases. The initial instruction of the reward models is done using the approximation of the highest probability. After that, all reward systems are employed for deduction, and the generation model is trained using the melodic ragas identified by the Bi-GRU model for a discount factor value of 0.5. The resulting model's weights are initialised using the pre-trained reward model with an approximate learning rate of 0.01. The input and improvised music samples are presented in Fig. 3.

After training the model for several epochs with different learning rates, the music improvisation exhibited by the model is evaluated using specific parameters. The parameters considered for the evaluation of music improvisation in this work include pitch frequency (PF), standard pitch delay (SPD), average distance between peaks (ADP), note duration gradient (NDG) and pitch class gradient (PCG).

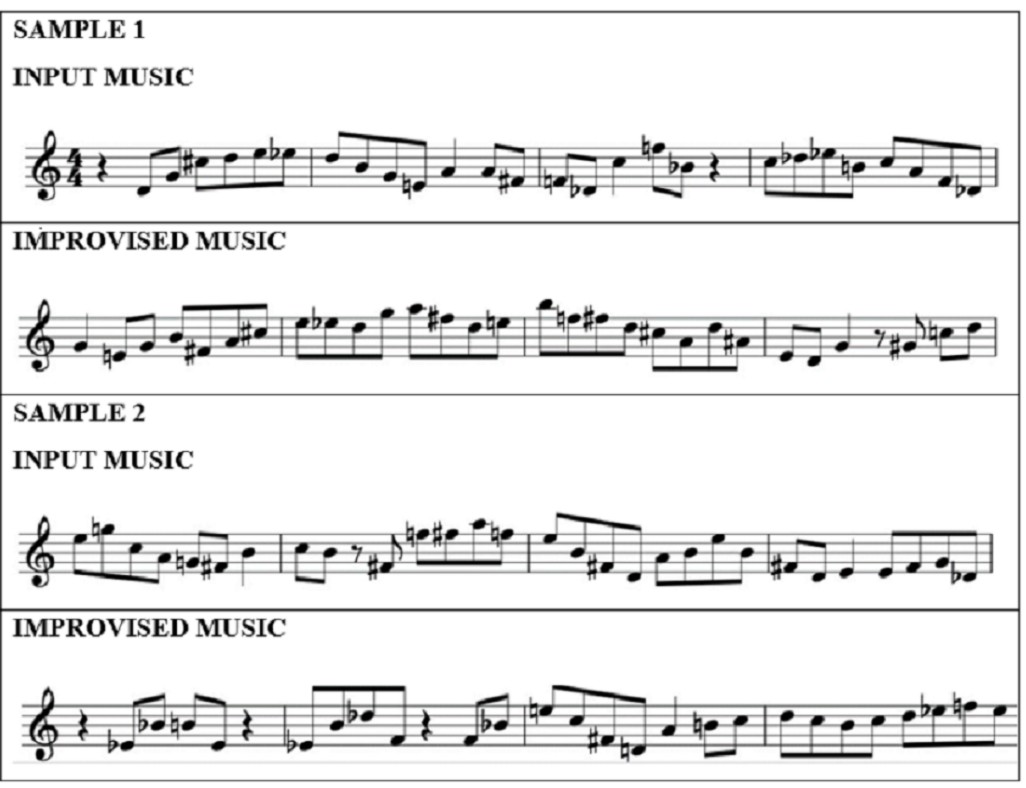

**Figure 3 Sample musical notes.**

**Table 2 Parameter assessment of music improvisation on various reinforcement methods.**

| Techniques | PF | SPD | ADP | NDG | PCG |
|---|---|---|---|---|---|
| Model based RL (*Keluskar et al., 2022*) | −0.86 | +3.45 | +0.94 | 0.0082 | 0.067 |
| Rule based RL (*Gurbuz-Dogan et al., 2021*) | −0.92 | +4.21 | +1.67 | 0.0097 | 0.072 |
| SARSA (*Rahman et al., 2021*) | +0.87 | −2.35 | −1.18 | 0.0074 | 0.053 |
| PPO (*Moreira, Furtado & Nogueira, 2020*) | +0.78 | −2.05 | −0.95 | 0.0062 | 0.048 |
| TRPO (*Surana, Varshney & Pendyala, 2022*) | +0.43 | −1.17 | −0.46 | 0.0053 | 0.036 |
| Deep Gradient based RL | +0.15 | −0.43 | −0.07 | 0.0041 | 0.025 |

Table 2 shows the values of the parameters obtained for various reinforcement learning-based algorithms.

The reinforcement learning methods such as the model-based RL algorithm, rule-based RL algorithm, State-Action-Reward-State-Action (SARSA) algorithm, proximal policy optimisation (PPO), and Trust Region Policy Optimization (TRPO) are taken for performance analysis of the approach for music improvisation in the proposed system. The

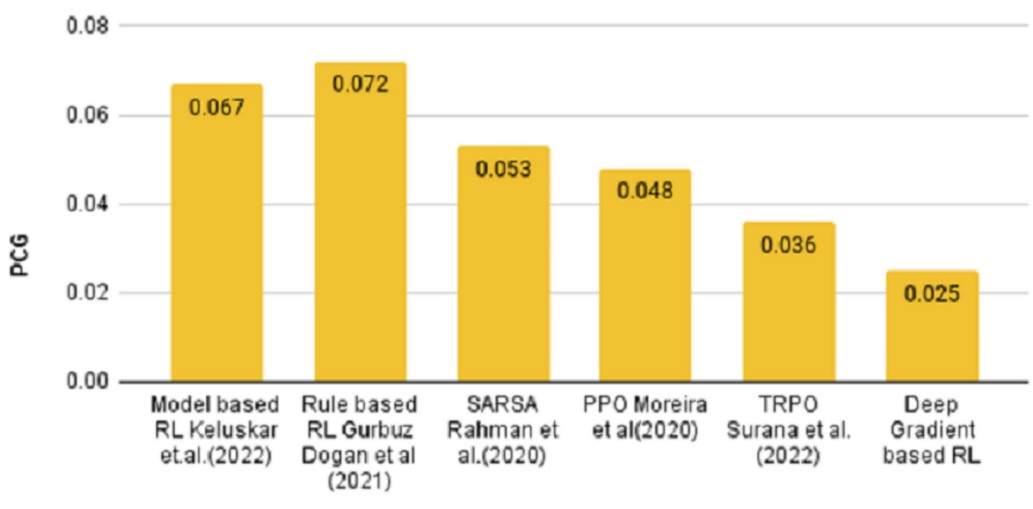

**Figure 4** **Performance comparison on NDG parameter performance with various RL methods** *vs* **proposed deep gradient RL method.**

pitch frequency of the model-based and rule-based algorithms lies in the negative notes, whereas for SARSA, PPO and TRPO algorithms, pitch frequency lies in positive notes, with DGBRL having the most negligible value.

On the contrary, the SPD value is positive for model and rule-based RL algorithms and stays in negative notes for SARSA, PPO, TRPO and DGBRL algorithms. The standard pitch delay is found to be lesser for the DGBRL technique. The average distance between the peaks is also less for DGBRL, with a −0.07 value, and the highest for the rule-based RL technique, with +1.67. Further, the values of two important gradients are also investigated. The values of NDG and PCG are minimal for the DGBRL method, with 0.0041 and 0.025. The NDG and PCG values are high, with 0.0082 and 0.067 for the model-based RL algorithm; however, they are lesser than those obtained for the rule-based RL algorithm. Figures 4 and 5 provide the graphical representations of the parameter assessments of NDG and PCG values.

Figure 6 compares the frequency and time domains of the AI-enhanced and original versions, showing both pattern similarities and differences. Initially, the time domain was slightly delayed in training, but there was no high variation between the original and improvised datasets.

## CONCLUSION

This work proposes music improvisation using BI-GRU and deep gradient-based Reinforcement learning. The melodic framework detected from input musical compositions using Bi-GRU is given as input to the DGBRL model. The improvised music is presented

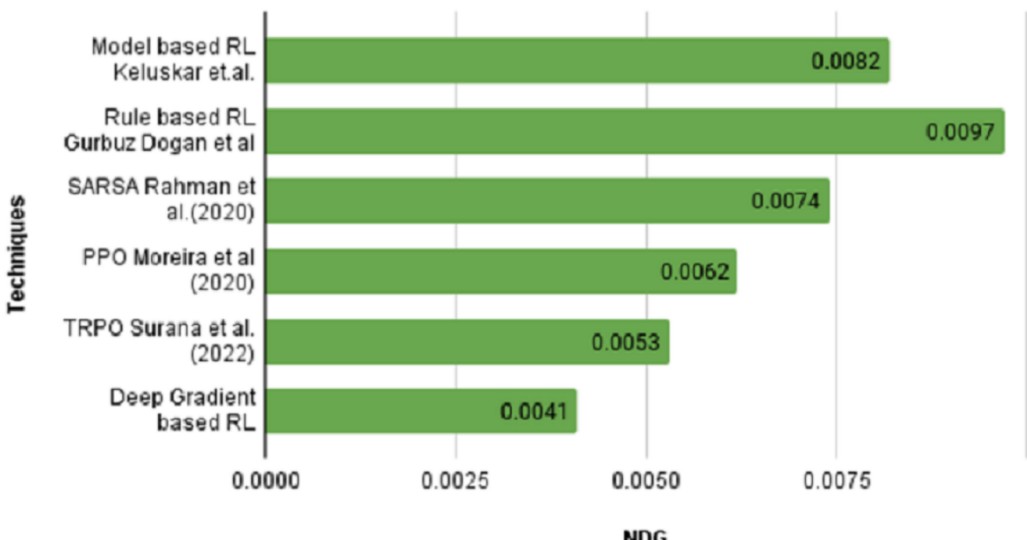

Figure 5   Performance comparison on PCG parameter.

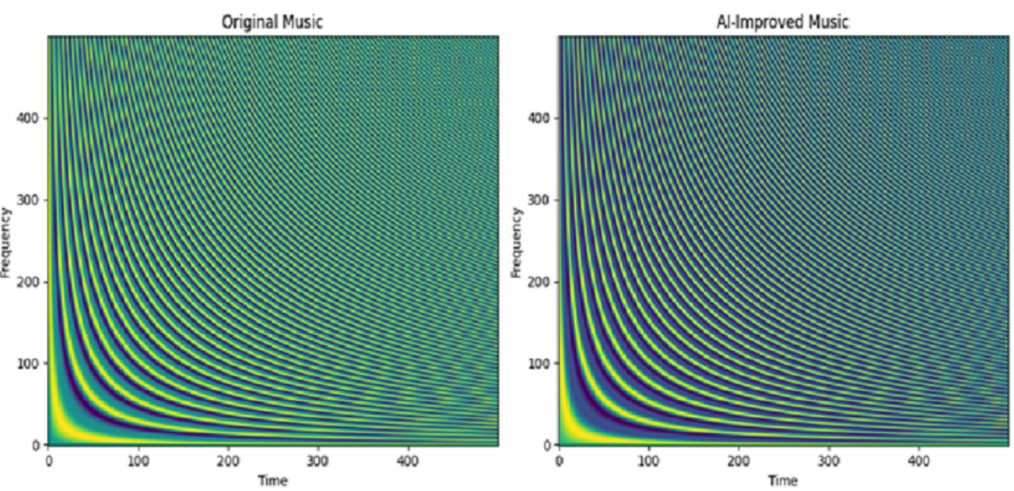

Figure 6   Spectrogram visualization for proposed music improvisation model for Dataset 2.

in MIDI format in the final output. By training on a more varied dataset, the proposed model can be improved in the future to analyse other musical genres and instrumentation.

The parameters like pitch frequency (PF), standard pitch delay (SPD), average distance between peaks (ADP), note duration gradient (NDG) and pitch class gradient (PCG) are compared on various reinforcement models on music improvisation in this work. The SPD value is positive for existing rule-based RL algorithms and stays in negative notes for SARSA, PPO, TRPO and DGBRL algorithms. On the other hand, in the proposed study,

the standard pitch delay is found to be lower for the DGBRL technique. The rule-based RL approach has the most significant distance between peaks (+1.67), whereas DGBRL has the lowest average distance between peaks (−0.07). This shows that our model accuracy on improvisation with similar notes is higher than that of other models. Nevertheless, our method has a few limitations. Due to its small training dataset, the model may not be able to generalize over a wide range of instrumentations and musical genres. Subsequent efforts may concentrate on several aspects to enhance the functionality and efficiency of the model. The ability of the model to assess and improvise across a variety of musical styles may be improved by training it on a more diverse and extensive dataset that includes various raga-related music excerpts. Another exciting application for this technology is the identification and improvisation of melodic frameworks in real-time during live performances.

In the future, Transformer models, whose attention mechanisms have demonstrated remarkable performance in managing sequence data, could greatly enhance the model's comprehension and production of intricate musical structures. Transformers are more effective than GRUs or LSTMs at capturing long-range dependencies in music. Transformer models could significantly improve the model's ability to understand and generate complex musical compositions.

### Funding
This work was supported by the Deanship of Scientific Research at King Khalid University through large group Research Project under grant number (RGP2/13/45), Princess Nourah bint Abdulrahman University Researchers Supporting Project number (PNURSP2024R77), Princess Nourah bint Abdulrahman University, Riyadh, Saudi Arabia, and the Prince Sattam bin Abdulaziz University project number (PSAU/2024/R/1445). This study is funded by the Future University in Egypt (FUE). The funders had no role in study design, data collection and analysis, decision to publish, or preparation of the manuscript.

### Grant Disclosures
The following grant information was disclosed by the authors:
Deanship of Scientific Research at King Khalid University: RGP2/13/45.
Princess Nourah bint Abdulrahman University Researchers Supporting Project: PNURSP2024R77.
Prince Sattam bin Abdulaziz University project: PSAU/2024/R/1445.

### Competing Interests
The authors declare there are no competing interests.

### Author Contributions
- Fadwa Alrowais conceived and designed the experiments, performed the computation work, prepared figures and/or tables, and approved the final draft.

- Munya A. Arasi conceived and designed the experiments, performed the computation work, prepared figures and/or tables, and approved the final draft.
- Saud S. Alotaibi conceived and designed the experiments, analyzed the data, prepared figures and/or tables, and approved the final draft.
- Mohammed Alonazi performed the experiments, analyzed the data, authored or reviewed drafts of the article, and approved the final draft.
- Radwa Marzouk performed the experiments, analyzed the data, performed the computation work, authored or reviewed drafts of the article, and approved the final draft.
- Ahmed S. Salama performed the experiments, analyzed the data, performed the computation work, authored or reviewed drafts of the article, and approved the final draft.

## Data Availability

The Bach Choarles dataset is available at UC Irvine:

Conklin, Darrell. Bach Chorales. UCI Machine Learning Repository. https://doi.org/10.24432/C5GC7P.

The code is available as a Supplemental File.

## Supplemental Information

Supplemental information for this article can be found online at http://dx.doi.org/10.7717/peerj-cs.2265#supplemental-information.

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
