# Peer review of "Deep gradient reinforcement learning for music improvisation in cloud computing framework"

_PeerJ Computer Science, doi:10.7717/peerj-cs.2265_

## Round 0.1 · original submission · Major Revisions

Dear Authors,

After carefully considering the reviews and assessing your manuscript, I am pleased to inform you that we would like to invite you to revise and resubmit your manuscript for further consideration. The reviewers have provided constructive comments that will help strengthen your work. Please address each of these points thoroughly in your revised manuscript. Additionally, ensure that you provide a detailed response letter outlining how you have addressed each comment raised by the reviewers. This will help the reviewers and myself to evaluate the changes made to the manuscript.

Reviewer 1 ·

Basic reporting

This article explores using reinforcement learning (RL) in music improvisation to enhance AI-human creative interactions. It outlines using RL to generate real-time musical compositions by training an RL agent with input data processed using Bi-directional Gated Recurrent Units. The agent's reward system, based on deep Gradient-based Reinforcement learning, guides it to create aesthetically pleasing and harmonically coherent music, rendered in MIDI format. The study uses the Bach Chorales dataset and evaluates improvisations based on parameters like pitch frequency, pitch delay, and note duration gradient.

General approach of the study and algorithms used are well thought out, but paper have some deficiencies. Most important problem is the language of the paper. There is a problem with English. For example, use of THE is excessive, it appears in many unnecessary places. I think google translate was used. It should definitely be fixed.

Experimental design

General organization of the paperis good, but following deficiencies should be corrected;
1) All Figures must be redrawn. Their resolution is very poor and not understandable.
2) Why was only accuracy rate used? Other performance metrics should also be evaluated.
3) Why has no study from 2023 been examined? This should definitely be added to literature section. It is like, the study was completed in 2022 and, it seems that it was previously rejected from other places and was never revised afterwards.

Validity of the findings

1) Figures should definitely be redrawn.
2) ROC/AUC values are important to us, their graphs should also be added.
3) Discussion and conclusion part is very inadequate. Conclusion part should be examined in more detail. If possible, Discussion section should be added.

Reviewer 2 ·

Basic reporting

In abstract it could benefit from a more concise presentation of the key findings.
Elaborate on the criteria used for selecting the Bach Chorales dataset and its relevance to your research objectives.
Discuss the scalability and performance implications of cloud and containerization architecture

Experimental design

Discuss potential future directions for improving the proposed model.
Include a brief discussion on the computational resources required for training and deploying your model in a cloud environment.
Images are not clear
include contribution statement.
Include motivations statement before literature review
Provide comaparative analysis tables with the benchmark models/exisitng models and discuss how your approach is outperforming them.
English and grammar needs improvements.
Overall manuscript needs improvement. May address all the above points

Validity of the findings

Discuss the generalizability of your findings beyond the Bach Chorales dataset and how your approach could be applied to diverse musical genres.
What is the significance of Bi-GRU and DGBRL in the context of music improvisation
Include visualizations or examples of the improvised music output. This will help readers better understand the quality and aesthetic appeal of the generated compositions.
Manuscript is not as per the format. Kindly take care of it.

Additional comments

no comment

---

## Round 0.2 · Minor Revisions

Some minor changes are required before manuscript accepted for publication.
1) Improve the language of the manuscript,as suggested by one of the reviewers.
2) Abstract needs to be modified. Beginning the sentence with "To foster... " Is not appreciate. Authors should begin the statement with the problem statement, motivation, etc. Also mention the obtained results in terms of used metrics such as PF, SPD, APD, etc
3) Legends of Table 1, Table 2, anf Fig 2-5 are short and it is very hard to follow. Write these appropriately, and also explain sufficiently within text.
4) Table 2, and Fig. 4-5 present comparison of the proposed method with other models. However, reference to published methods used for the comparison are not mentioned in Table 2 and Figs.
5) Conclusion is very tightly written. Authors must conclude paper appropriately, mention limitations and limitations, followed by future work Authors mentioned Transformer model here, but should also explain appreciate reason.

Looking forward to receiving revision. Good luck.

Reviewer 2 ·

Basic reporting

English needs improvements.

Experimental design

no comment

Validity of the findings

no comment

Additional comments

no comment

---

## Round 0.3 · accepted · Accept

I am pleased to inform you that your paper has been accepted for publication in PeerJ Computer Science. Your manuscript has undergone rigorous peer review, and I am delighted to say that it has been met with praise from our reviewers and editorial team. Your research makes a significant contribution to the field, and we believe it will be of great interest to our readership. On behalf of the editorial board, I extend our warmest congratulations to you.